# Associations Between Uraemic Toxins and Gut Microbiota in Adults Initiating Peritoneal Dialysis

**DOI:** 10.3390/toxins17070334

**Published:** 2025-07-01

**Authors:** Philippa James, Jordan Stanford, Ojas V. A. Dixit, Mary Ann Nicdao, Brett McWhinney, Kamal Sud, Michele Ryan, Scott Read, Golo Ahlenstiel, Kelly Lambert, Claire O’Brien, Katrina Chau

**Affiliations:** 1School of Medicine, Western Sydney University, Campbelltown, NSW 2560, Australia; 2School of Medical, Indigenous and Health Sciences, Faculty of Science, Medicine and Health, University of Wollongong, Wollongong, NSW 2522, Australia; 3Faculty of Science and Technology, University of Canberra, Canberra, ACT 2617, Australia; 4Western Renal Services, Western Sydney and Nepean Blue Mountains Local Health Districts, Sydney, NSW 2750, Australia; 5Analytical Chemistry Unit, Chemical Pathology, Pathology Queensland, Brisbane, QLD 4029, Australia; 6Nepean Clinical School, Faculty of Medicine and Health, The University of Sydney, Sydney, NSW 2050, Australia; 7Liver Immunology Group, Storr Liver Centre, Westmead Institute for Medical Research, Westmead, NSW 2145, Australia; 8Blacktown-Mount Druitt Hospital, Blacktown, NSW 2148, Australia; 9Health Innovations, University of Wollongong, Wollongong, NSW 2500, Australia; 10Kidney Lifestyle Research Group, University of Wollongong, Wollongong, NSW 2500, Australia

**Keywords:** chronic kidney disease, kidney failure, gut microbiome, uraemic toxin, peritoneal dialysis, diet therapy

## Abstract

Declining kidney function contributes to the accumulation of uraemic toxins produced by gut microbiota, leading to the uraemic syndrome. This study aimed to identify associations between uraemic toxins, diet quality, symptoms and the gut microbiota in individuals initiating peritoneal dialysis. A cross-sectional analysis of baseline data from participants in a longitudinal study was conducted. Symptom scores using the Integrated Palliative Care Outcomes Scale-Renal were recorded. Plasma p-Cresyl sulfate, indoxyl sulfate and trimethylamine N-oxide were measured using liquid chromatography-mass spectrometry. Gut microbiota was determined using 16S rRNA sequencing. Multivariate linear models examined associations across the cohort. Data from 43 participants (mean age 61 ± 13 years; 70% male; median eGFR 7 mL/min/1.73 m^2^) were analysed. Diabetes was the primary cause of kidney disease (51.2%). Patients were classified into ‘high’ (*n* = 18) and ‘low’ (*n* = 26) uraemic toxin groups using K-means clustering. The ‘high’ group had a lower eGFR (*p* < 0.05) but no differences in diet quality or symptom scores. Significant differences in alpha and beta diversity were observed between the groups (*p* = 0.01). The ‘high’ group had increased *Catenibacterium*, *Prevotella*, *Clostridia*, and decreased *Ruminococcus gnavus* abundances. Multivariate models identified 32 genera associated with uraemic toxins, including positive associations of *Oscillospiraceae* UCG-002 and UCG-005 with p-cresyl sulfate, and negative associations with *Actinomyces* and *Enterococcus*. Patients with kidney failure initiating peritoneal dialysis have distinct uraemic toxin profiles, associated with differences in microbial diversity. This phenotype was also associated with differences in residual kidney function but not with diet or symptom severity. Longitudinal studies are required to determine causality and guide therapeutic interventions.

## 1. Introduction

Chronic kidney disease (CKD) is a highly prevalent disorder that results in a major burden of disease worldwide [1]. Advanced CKD manifests as the uraemic syndrome, a nebulous cluster of symptoms including fatigue, pain and pruritus [2,3]. This syndrome is thought to develop due to the accumulation of uraemic toxins, of which over 100 are identified [3]. Uraemic toxins, such as P-cresyl sulfate (PCS), indoxyl sulfate (IS) and trimethyl-amine-N-oxide (TMAO), accumulate via a combination of decreased renal clearance and increased production by gut microbes [4]. There is a significant increase in uraemic toxin serum concentration, including PCS and IS, as the estimated glomerular filtration rate (eGFR) decreases [5]. Individuals with CKD have been found to have distinct gut microbiota composition and function, including an increased abundance of *Enterobacteria* and *Enterococci* species, and lower numbers of *Bifidobacterium* species [6,7,8]. Mechanisms contributing to dysbiosis (an imbalance in microbial composition) in individuals with CKD include the increased concentration of urea in the gut lumen, which exerts a selective pressure favouring the growth of bacteria expressing urease and enzymes capable of forming indole and p-cresyl [9,10]. Dietary changes utilised in CKD management, such as low potassium and low phosphate diets, may further exacerbate dysbiosis; this is also the case for iron supplementation, phosphate binders, volume overload and metabolic acidosis [2,9,10,11,12]. Uraemic toxins are thought to result in clinical symptoms [13,14], CKD progression [15], and increased risk of death [16,17]. While dialysis is utilised to supplement kidney failure, it often only attenuates the uraemic syndrome and patients are left with residual symptoms [4,18]. This might be due to dialysis having limited efficacy in removing uraemic toxins from the body because of large molecular size, protein binding and sequestration within body compartments [18,19].

Given the adverse effects of the uraemic toxins PCS, IS and TMAO, and the limited efficacy of dialysis in clearing them from the patient’s body, the potential for dietary modification to alter gut microbiota composition and thereby minimise serum uraemic toxin levels has been hypothesised. Diet has been shown to play a role in modulating the composition and metabolic function of human gut microbiota [12,20]. However, there are few studies that have explored the effects of specific foods and nutrients on the gut microbiome and uraemic toxins in a CKD stage 5 population or patients undergoing treatment with peritoneal dialysis (PD). There is significant potential to develop treatments, including diet modification and probiotic therapies, that influence gut microbiome composition and thereby reduce the production of uraemic toxins. In turn, it is expected that the patient would experience a reduction in uraemic symptom burden, slower CKD progression, and fewer complications from their disease.

This study aimed to explore associations among plasma uraemic toxins, the gut microbiota profile, diet, and patient-reported symptoms in adults with kidney failure at the commencement of PD. This is the baseline observational data in a longitudinal study of the microbiome in patients undergoing PD. In particular, we wanted to determine whether elevated uraemic toxins were associated with higher patient-reported symptoms, differences in dietary quality and gut microbial profile, enabling the targeted development of potential therapies to modify uraemic toxin levels.

## 2. Results

### 2.1. Participant Characteristics

A total of 44 participants enrolled in the study within the first 3 days of commencing PD. However, 43 had completed data, including uraemic toxin data (Table 1), which formed the analytical dataset for this study. Of these individuals, 30% (*n* = 13) were females, with a mean age of 61 ± 13 years. Baseline renal function ranged from eGFR of 3 to 13 mL/min/1.73 m^2^. The most frequent cause of renal failure was diabetes mellitus (*n* = 22, 51.2%). Of the 43 participants, 88.4% (*n* = 38) had taken antibiotics in the preceding 12 months (Table 1). Sample numbers analysed per outcome varied (Appendix A). This resulted from patients’ reluctance to participate in aspects of the study, particularly diet history and stool sampling. Median serum concentrations of the uraemic toxins TMAO, total and free IS, and total and free PCS were above the reference range of healthy controls with a wide range of serum toxin levels [21,22]. Serum concentrations of uraemic toxins are summarised in Table 1. Additional biochemical data are in Appendix A. Baseline dietary data is summarised in Table 2.

### 2.2. Uraemic Symptoms

Patients reported numerous symptoms, the most frequent of which was weakness or lack of energy (*n* = 32, 74%). The other most reported symptoms in order of frequency were constipation (*n* = 31, 72%), pruritus (*n* = 29, 67%) and difficulty sleeping (*n* = 28, 65%) as seen in Figure 1.

### 2.3. Uraemic Toxin Phenotype

To explore disparities in baseline characteristics between subgroups with higher or lower uraemic toxin levels, we employed k-means clustering to categorise participants by baseline levels of all five uraemic toxin measures (Figure 2). One distinct cluster, termed the “high uraemic toxin group” (high), *n* = 18, exhibited consistently elevated uraemic toxin concentrations relative to the “low uraemic toxin group” (low), *n* = 25, which displayed lower levels of one or more uraemic toxins.

There was no difference in the severity of total symptom scores between the high and low uraemic toxin groups (Figure 3A). Examining the relationship between the presence or absence of symptoms and plasma uraemic toxin levels using binary logistic regression while adjusting for age, gender, albumin, and diabetes did not reveal any statistically significant associations.

With regards to dietary intakes, no significant difference was found between the high and low uraemic toxin groups for plant-based diet quality indices or selected nutrients including protein, fibre and their ratio (Figure 3B), nor between individual uraemic toxins and daily nutrient intakes (Appendix A).

### 2.4. Uraemic Toxin Phenotype and the Gut Microbiota

Given the role of gut microbiota in the production of uraemic toxins, we examined the microbiome of the high and low uraemic toxin groups to identify differences in microbial composition. A significant difference in alpha diversity between the two groups was observed, with the high uraemic toxin group displaying higher alpha diversity indices: pPielou = 0.01, pShannon = 0.01, pSimpson = 0.03, pFaith = 0.01 (Figure 4A). There was also a significant difference in beta diversity between the high and low groups (*p* = 0.01, R^2^ = 0.32), indicating that 32% of the variation in microbial community composition can be attributed to differences between these groups (Figure 4B).

Seven individual taxa showed significant differences between the cluster groups (Table 3). Of these, six were more abundant in the high uraemic toxin group, while *Ruminococcus gnavus* was the only taxa found in higher abundance in the low toxin cluster (Table 3). The relationships between individual taxa, uraemic toxin concentrations, and total symptom scores were explored (Figure 5). *Oscillospiraceae* UCG-002 and UCG-005 were positively associated with total and free PCS, whereas *Actinomyces* and *Enterococcus* exhibited negative associations (Figure 5).

Furthermore, higher symptom scores were negatively associated with *Caproiciproducens* (adjusted *p* = 0.02), *Paraprevotella* (adjusted *p* = 0.02), and *Family XIII AD3011* (adjusted *p* = 0.02), while positive associations were observed with *Acetanaerobacterium* (*p* = 0.05) and *Clostridia innocuum* (*p* = 0.05).

The log_2_ fold change represents the difference in abundance between the high and low uraemic toxin cluster groups. A log_2_ fold change of 1 indicates that a taxon is twice as abundant in the high uraemic toxin group compared to the low group, while a log_2_ fold change of −1 means it is half as abundant in the high uraemic toxin group than the low group. Data were generated using DESeq2 [25], based on count abundance data at the genus level. Data were rounded to two decimal places. Analysis was conducted based on completed microbiota data from *N* = 34 participants.

## 3. Discussion

This observational study of patients with kidney failure has revealed complex relationships between uraemic toxins, uraemic symptoms, diet and gut microbiota. In our study, the profile of plasma uraemic toxins could be classified into ‘high’ and ‘low’ toxin phenotypes, which were related to residual renal function. While these clusters did not identify differences in uraemic symptoms or dietary patterns, they were associated with differences in alpha and beta diversity of the gut microbiome.

Alpha diversity measured using multiple indices was significantly elevated in the high uraemic toxin cluster. This appears paradoxical given that increased diversity is generally assumed to be associated with health and is not expected to be increased with high uraemic toxin concentrations [26,27]. However, this parameter may reflect the proliferation of opportunistic bacteria that thrive in this environment [28]. While *Clostridia*, *Lachnospiraceae*, and *Eubacterium*—which have been shown in in vitro studies to produce p-cresol, the precursor of PCS [29]—were more abundant in the high uraemic toxin group, this group also had lower residual renal function, potentially limiting uraemic toxin excretion. This makes it difficult to determine whether the relationship between these taxa and uraemic toxins is causal or simply a consequence of impaired kidney function.

Previous studies have shown that CKD patients have reduced alpha diversity, but these studies compared patients with CKD to normal populations rather than within a CKD stage. For example, a study of alpha diversity in 95 patients with newly diagnosed CKD of stage 1 to 5, showed that in comparison to 20 healthy controls alpha diversity measured by Chao1 and ACE diversity indices was decreased in the CKD cohort, but the Simpson’s and Shannon’s indices did not vary significantly [30]. Liu et al. also analysed 315 gut microbiota samples from patients with CKD and healthy controls, showing that while the Shannon index was significantly altered, the ACE index was not, inferring that CKD mainly affected the evenness (how species are distributed) of gut microbiota rather than richness (number of different species) [31].

Our findings contribute to the results of previous work as we identify significant differences in alpha diversity even within a narrow range of residual renal function in patients deemed to require commencement of PD. While the interpretation of alpha diversity is dependent on the index used, our results showed a consistent increase across multiple indices, suggesting increases in both richness and evenness. With regard to specific taxa, previous studies have shown reduced abundance of genus *Prevotella* in CKD [32], but our data show a significant difference in this genus between the high and low clusters. This builds on previous data showing correlations between specific bacteria, uraemic toxins and CKD progression. Faecal samples of CKD patients demonstrate lower relative abundance of short-chain fatty acid generating bacteria, such as *Bifidobacterium* spp., and higher relative abundance of *Enterobacteriaceae* in CKD [33]. In rat models, *Fusobacterium nucleatum* and *Eggerthella lenta* increase uraemic toxin production [33]. A recent study has also demonstrated the association between reduced abundance of *Lactobacillus johnsonii* and CKD progression, with *L. johnsonii* supplementation attenuating renal injury [34]. Our study also demonstrated that the high and low uraemic toxin groups had significantly different beta-diversity. Considering that the difference in residual renal function between the two groups was merely 2 mL/min, this suggests that the microbial community can shift significantly even within a CKD stage. The time frame for the reduction in eGFR at this stage of CKD varies significantly from months to years. Our results suggest that a more nuanced approach to microbiome analysis may be required in the CKD population. Future work focusing on gut microbiome characteristics and the rate of decline in kidney function may be warranted.

Despite these differences in the gut microbiota profile, our study found no difference in uraemic symptoms between the two clusters. While one would expect that the high uraemic toxin group would have more severe symptoms, our findings were consistent with previous work, which has demonstrated a similar lack of correlation. The EQUAL study was an observational cohort study of 795 patients with stage 4 or 5 CKD, not on dialysis [35]. These patients had their symptoms measured using the Dialysis Symptom Index [13]. Ten uraemic toxins were analysed, including those analysed in our study, but only TMAO was positively associated with fatigue and PCS with constipation [13]. This was only demonstrated in men, possibly due to the higher serum toxin concentrations in males [13].

The association between the specific symptom of pruritus and uraemic toxins is similarly conflicting. Wang et al. studied 320 patients with CKD who had an eGFR of between 10 and 99 mL/min and measured their experience of pruritus with a visual analogue scale and questionnaire [14]. The concentration of total PCS was positively associated with pruritus [14]. This finding was not reproduced by Świerczyńska-Mróz et al., who assessed pruritus severity using a numerical rating scale, and the validated instruments UP-Dial, ItchyQoL and the Four-Item Itch Questionnaire in 124 haemodialysis patients with and without pruritus, as well as 50 healthy controls [36]. There was no significant difference in the uraemic toxin levels between patients with or without pruritus [36]. The explanation for the lack of association between uraemic toxins and uraemic symptoms remains unclear. It is known that the level of uraemic toxins and symptoms are elevated in the later stages of CKD, and the consensus is that they are related. However, a causal pathway has not been identified. It is hypothesised that there may be other uraemic toxins that are yet unstudied, as well as other physical, psychological, or social factors that are also contributing to how symptoms are perceived and reported.

Our data showed a significant association between specific microbial taxa and total symptom burden (iPOS score). The most significant associations were a negative for *Caproiciproducens*, *Paraprevotella* and *Family XIII AD3011* and a positive for *Acetanaerobacterium* and *Clostridia*. Of these, the genus *Paraprevotella* has been found to be correlated with schizophrenia and depression [37], and *Acetanaerobacterium* associated with bipolar disorder [38]. The relationship between these organisms and symptoms remains unclear, but further work on the gut–brain axis may provide clarity in the future.

Our research contributes to the understanding of interactions between diet and uraemic toxins [39]. In our cohort, neither cluster type nor individual toxins were associated with dietary nutrients. Previous analyses show that increased dietary fibre intake is associated with lower serum uraemic toxin levels. A meta-analysis by Yang et al. identified ten randomised controlled trials involving 292 patients with CKD, in which dietary fibre supplementation significantly reduced levels of both IS and PCS [40]. The results of subgroup analyses suggested that probiotics/synbiotics were more effective in reducing BUN in non-hemodialysis CKD patients. This is probably due to the fact that CKD in hemodialysis patients usually has progressed to the stage of end-stage renal disease, and the renal units are irreversibly damaged, which makes it difficult to improve by drug treatment [41]. Wu et al. also performed a meta-analysis of data from experimental studies comprising a total of 203 patients with stage 3 to 5 CKD [42]. There was no effect of fibre supplementation on plasma IS for CKD patients, but the PCS level was significantly reduced [42]. Overall, these meta-analyses show significant interstudy variability in patient characteristics (inpatient vs. outpatient setting, age, stage of CKD, ethnicity, type of dialysis) and in type, dose and duration of fibre supplementation. The types of dietary fibre supplementation in these studies included non-starch polysaccharides, resistant oligosaccharides, resistant starch, and mixed forms of dietary fibre. The dose ranged from 6 g/day to 50 g/day, for a duration of 15 days to 16 weeks [40,42,43]. In our study, the dietary intake of fibre was 23.8 g/day. The geographical, cultural and genetic differences in populations, as well as differences in the types, doses, and durations of fibre intake, may explain the differences in the results of these studies.

These conflicting findings may also be explained by the composition of the overall diet. Rossi et al. analysed PCS and IS in 40 CKD patients with eGFR 24 +/− 8 mL/min/1.73 m^2^ and found that dietary fibre was positively associated with free and total PCS, but not with IS [44]. However, the protein–fibre ratio was positively associated with both total serum PCS and IS, leading them to conclude that the interplay between these nutrients is important [44]. This association between protein–fibre ratio and uraemic toxins was not noted in our analyses, and our study did not demonstrate an association between more robust measures of overall diet composition, specifically plant-based diet indices (PDI), with uraemic toxin levels. In recent years, research into plant-based diets has further divided the PDI into overall PDI, as well as healthful PDI (hPDI) and unhealthful PDI (uPDI) [45]. Healthful PDI places emphasis on the intake of whole grains, fruits and vegetables, and uPDI is characterised by the intake of refined grains and sugar-sweetened beverages [45]. McFarlane et al. have demonstrated a correlation between hPDI and lower levels of free PCS [46]. Similarly, Stanford et al. showed that uPDI was associated with increased free and total IS while hPDI was negatively associated with total serum IS concentration [28]. However, our cohort had more advanced kidney disease, and reduced excretion of uraemic toxins may overwhelm any effect that diet has on toxin production. Associations between diet and gut microbiota-derived uraemic toxins were further elucidated by Czaja-Stolc et al., who conducted an observational, cross-sectional study in adult dialysis patients and kidney transplant recipients (84 patients on haemodialysis, 44 PD patients, 54 kidney transplant recipients and 30 healthy controls) [47]. Similar to our research, PCS, IS and TMAO were measured, and a wide range in uraemic toxin concentration was observed [48]. An alternative dietary measure, the aMED score, was calculated from analysis of participants’ 3-day food diaries [48]. Dialysis patients who more closely followed a Mediterranean diet had significantly lower PCS concentrations [48]. Diets higher in vegetables were shown to significantly reduce the effect of phenylalanine and tyrosine intake on PCS levels in dialysis patients [48]. The study suggests that higher vegetable intake mitigates against the effect of protein intake on PCS levels, although there was no significant correlation between protein/fibre ratio demonstrated [48]. Considering these varying findings, we hypothesise that dietary approaches to reducing uraemic toxins likely require personalisation.

The strength of this study lies in its methodology. Data collection, including antibiotic and medication history and detailed dietary history and analysis performed by a specialist renal dietician, ensured greater accuracy than recall surveys. Further, analysis of dietary indices in addition to individual nutrients allows for real-world applicability, given that people eat whole foods. The uraemic toxins included in the study were selected for their demonstrated roles in CKD progression and cardiovascular mortality. Despite these strengths, the study was limited by the small population and short length of the dietary record. There is also the possibility of type 1 errors, although a robust statistical analysis was performed to address this.

A limitation of this study is its cross-sectional design, which restricts the ability to infer causal relationships and limits the scope to associations only. The effect of peritoneal dialysis on parameters analysed in this study also remains to be determined in forthcoming results in our longitudinal study. Additionally, the study included limited data on microbial metabolites and their potential effects. Although technological advances continue to emerge in this field [49], this study did not incorporate comprehensive metabolomic or metagenomic techniques, which may have enabled deeper. Another limitation in our study cohort is the pervasive use of antibiotics such as those for prophylaxis of PD catheter-related infections. While this poses challenges for microbiome-related investigations, it nonetheless reflects real-world clinical practice. Medication use is often not reported in studies of the microbiome in CKD. Therefore, comparison with other studies is difficult. As noted by Krukowski et al., covariates known to affect the gut microbiome, including drug therapy, are often not recorded in microbiome studies of CKD patients, which could bias results and preclude appropriate de-confounder analysis [50].

This research adds to the body of evidence on the gut microbiome and uraemic toxins in patients with CKD. To our knowledge, profiles of uraemic toxins and their relationship with the microbiome have not previously been studied in patients at the commencement of PD. Even within a narrow range of residual renal function, patients could be classified into uraemic toxin profile phenotypes, which in turn revealed a difference in both alpha and beta diversity. Although these did not relate to differences in dietary composition or uraemic symptoms, this approach may be more useful in the personalisation of future interventions. Measuring common uraemic toxins may be a more readily available assay of downstream metabolic effects of an individual microbiome as compared to metabolomics and allow targeted pre-, pro-, and symbiotic therapies, which have currently displayed variable results. Uraemic toxins in and of themselves may be a worthy treatment target to reduce other toxicities, even if symptoms are not reduced, and this needs to be clarified with further work. Future directions that include metagenomics and gene catalogues, as well as metabolomics, may provide a way of breaking through to therapies with clinical significance.

## 4. Materials and Methods

This is a cross-sectional analysis of baseline data from a 12-month longitudinal study following adults before and after the commencement of PD. This study was approved by the Western Sydney Local Health District Human Research Ethics Committee (2019ETH12569).

### 4.1. Subjects

Patients attending the Regional Dialysis Centre at Blacktown Hospital for PD training between 1 June 2020 and 12 July 2021 were invited to participate in the study. Criteria for inclusion were the ability to give informed consent, being eighteen years or older, and having a known diagnosis of kidney failure. Patients were excluded if they were pregnant or lactating. The COVID pandemic impacted patient recruitment as resources were diverted to the pandemic response. Laboratory analyses were also delayed.

### 4.2. Sample and Data Collection

Patient specimens were collected. Patient blood was obtained on the first day of PD training, which is undertaken with the intention to continue PD as a long-term kidney replacement therapy. Blood and faeces samples were obtained from each participant. Non-fasting peripheral blood was collected into tubes (K2E BD Vacutainer 18 mg EDTA, BD-Plymouth, PL6 7BP, UK). Samples were centrifuged at 1600 g for ten minutes (Allegra X-12R, Beckman Coulter, Lane Cove, Sydney, NSW, Australia), plasma extracted and stored in 2 mL aliquots at −80 °C. Patients were provided with a kit containing instructions for the collection of the faecal sample and a Coloff bag to collect the stool. Study participants collected the sample at the Regional Dialysis Centre or at home during the training period (3 days). If collected at home, the stool was delivered to the centre within 24 h. The stool was then transferred to a DNA/RNA Shield Fecal Collection Tube (Zymo Research, Irvine, CA, USA). Faecal samples were stored at −80 °C within 24 h. DNA is stable in this collection tube for 2 years, and the RNA for 1 month at room temperature [47].

Symptom burden was ascertained using the Palliative Care Outcome Scale—Renal Version (iPOS-Renal). This questionnaire was completed via interview by an investigator for each participant.

Dietary data were collected by the Renal Dietitian (MR) on site at the Regional Dialysis Centre. Participants recorded in a written diary at least three days of eating. This was supplemented with photographs of all meals and snacks eaten for at least one day to verify intake and portion sizes. Instruction was provided for individuals unfamiliar with taking photos on their phone. A renal dietitian interpreted the photos and recorded each diet history in writing. If the photos provided insufficient information for dietary analysis, the dietitian clarified the information in person or by phone.

For each participant, the electronic medical record was examined for the period of the study and the preceding eighteen months. Data extracted included demographic information, primary renal disease, comorbidities, current medications, use of antibiotics in the preceding twelve months, baseline physical measurements, and baseline biochemistry data.

### 4.3. Sample and Diet Analysis

#### 4.3.1. Microbial DNA Analysis

Genomic DNA was extracted from stool using QIAamp Fast DNA Stool Mini Kit (QIAGEN^®^, Hilden, Germany). The quality and quantity of DNA extracts were analysed using a NanoDrop™ 3300 spectrophotometer (Thermo Scientific™, Waltham, MA, USA). PCR amplification was performed using 16S rRNA gene universal primers targeting the V3–V4 region of the bacterial 16S rRNA gene. The PCR product is indexed and purified using Nextera XT DNA library preparation kit (Illumina Inc., Cat #: FC131-1096, San Diego, CA, USA) and AMPure XP (Beckman and Coulter Inc., Cat #: A63880, Brea, CA, USA), then sequenced on an Illuminia MiSeq system (Illumina Inc., San Diego, CA, USA) in paired-end 300 bp format with a minimum expected 10 000 reads per sample. QIIME2 version 2022.2 was used to analyse the 16S rRNA analysis. The Silva classifier was used to determine taxa.

#### 4.3.2. Uraemic Toxin Analysis

Plasma PCS and IS concentrations were quantified using Ultra Performance Liquid chromatography (UPLC) coupled with Fluorescence detection (Waters Corporation, Milford, MA, USA). Plasma PCS and IS were chromatographed using a Waters Acquity HSS T3 1.8 µm column. Mobile phase A was 50 mmol/L ammonium formate, and mobile phase B was 100% acetonitrile. An injection volume of 2 µL was used for total PCS and total IS, and 5 µL for free PCS and free IS; PCS, IS and the internal standard (50 µmol/L 4-ethylphenol) were quantified with timed programmed fluorescence detection monitoring at specific excitation–emission wavelengths (PCS: 260/283 nm; IS 300/390 nm; 4-ethylphenol: 285/310 nm).

Plasma TMAO was measured using the Acquity UPLC system, MassLynx V4.2 software and the Xevo TQD mass spectrometer (Waters, Milford, MA, USA). Plasma (50 µL) was mixed with 200 µL of acetonitrile, containing deuterated internal standard and centrifuged for 10 min. The supernatant (1 µL) was injected onto the system and chromatographically resolved using a Waters Acquity UPLC BEH Amide column (1.8 µM, 2.1 × 100 mm). The mobile phases consisted of 100 mM ammonium formate, pH 3.0 (mobile phase A) and 100% Acetonitrile (mobile phase B) and were pumped at 0.4 ml/min. The column temperature was set to 45 °C. A linear gradient from 0–25% Mobile phase A was run for 2.5 min and then returned to initial conditions from 2.51 to 4 min.

Quantification was performed by mass spectrometry using external calibration and an isotopically labelled internal standard. The assay was linear from 0.1 to 200 μmol/L, and recovery was 98–102% in patient samples. The interday imprecision across four different QC levels was below 7% for TMAO.

#### 4.3.3. Dietary Analysis

Dietary data was entered manually into FoodWorks (2019, Xyris Foodworks Version 10, Brisbane, QLD, Australia) using the AUSNUT 2011-2013 Australian Food Composition Database. For meals with multiple ingredients where the recipe was not provided, recipes were obtained from websites accessed via Google. PDI were calculated using three a priori plant-based diet indices (overall PDI, hPDI, uPDI) [51]. To achieve this, the Australian plant-based database was used to quantify plant and animal intake from single-ingredient, multi-ingredient, and mixed dishes [51]. The total scores are as follows (from lowest to highest possible score): PDI (46–230); hPDI (53–265); uPDI (51–255). A higher PDI score represents greater consumption of all types of plant foods, regardless of healthiness, while a higher hPDI score reflects greater consumption of healthy plant foods, such as whole grains, fruits, vegetables and nuts and less of unhealthy plant foods, including refined grains and sugars [52]. Conversely, a higher uPDI score indicates lower intakes of healthy plant foods and higher intakes of unhealthy plant foods, particularly from discretionary sources [52]. Detailed information on the dietary analysis is published elsewhere [53].

### 4.4. Statistical Analysis

All statistical analysis was undertaken in R (version 4.3.1; R Core Team, Vienna, Austria). K-means clustering was applied to categorise participants based on their uraemic toxin profiles to identify subgroups with distinct serum uraemic toxin concentrations. One participant was excluded due to lost plasma samples, resulting in a final analytical sample of 43 participants. Clustering was performed using the ComplexHeatmap package in R [54], where data were scaled but not centred (scale, center = FALSE). The clustering analysis used Euclidean distance for row-wise clustering and Ward’s method (ward.D) to define cluster membership.

Microbial beta diversity was analysed using arcsine square root transformed relative abundance data. Taxa present in less than 20% of samples were removed before analysis. Bray–Curtis dissimilarity indices were computed using the vegdist function using the vegan package [55], followed by Principal Component Analysis (PCA) using the prcomp function with scaled and centred data to visualise differences in microbiota composition. To assess whether microbial composition statistically differed between uraemic toxin clusters, PERMANOVA was conducted using the adonis2 function part of the vegan package [55], adjusting for age, gender, and diabetes status, and was based on 999 permutations.

To identify differences in the relative abundance of individual microbial taxa between high and low uraemic toxin clusters, differential abundance testing was performed using the DESeq2 package [25]. Prior to analysis, taxa present in fewer than 20% of samples were excluded. DESeq2 models were fitted using raw count data at the genus level. Multiple testing corrections were applied using the package’s default settings.

Associations between microbial taxa relative abundance, uraemic toxin concentrations, and total symptom scores (iPOS-renal) were assessed using linear regression models, as were associations between uraemic toxin concentrations and dietary intake.

The relationship between symptoms and uraemic toxin levels was examined using binary logistic regression. Due to the small sample sizes across the Likert scale of symptom severity, we collapsed the POS-Renal symptom categories into binary variables. Logistic regression models were then used to assess the odds of symptom presence or absence in relation to plasma uraemic toxin concentrations, adjusting for age, gender, blood albumin levels, and the presence of diabetes.

Associations between microbial taxa relative abundance, uraemic toxin concentrations, and total symptom scores (iPOS-renal) were assessed using linear regression models. Plasma uraemic toxin concentrations were log-transformed before analysis. Covariates included age, gender, diabetes status, and blood albumin levels. Associations between uraemic toxin concentrations and dietary intake were also assessed using linear regression models; the above covariates were analysed together with daily caloric intake (kJ/day). Further detail is provided in Appendix B.

## Figures and Tables

**Figure 1 toxins-17-00334-f001:**
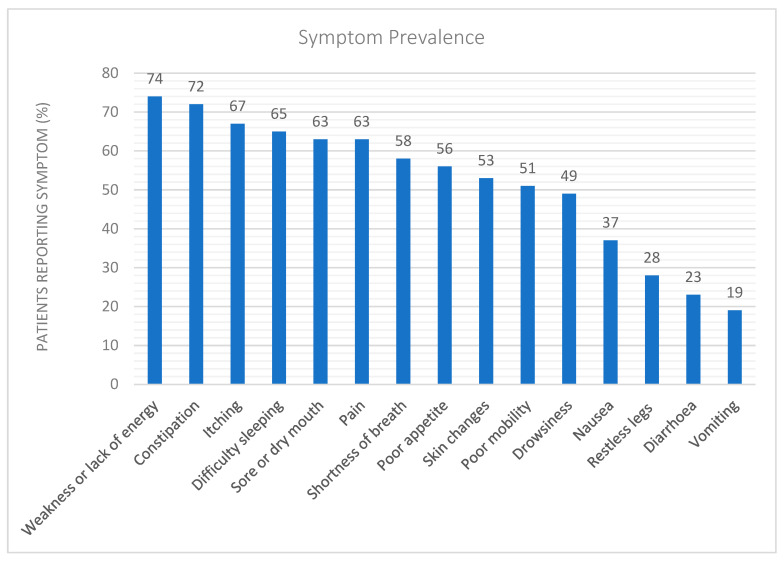
Prevalence of patient-reported symptoms. Physical symptoms associated with the uraemic syndrome were prevalent in this study population (*n* = 43). Most commonly reported symptoms were weakness or lack of energy, followed by constipation.

**Figure 2 toxins-17-00334-f002:**
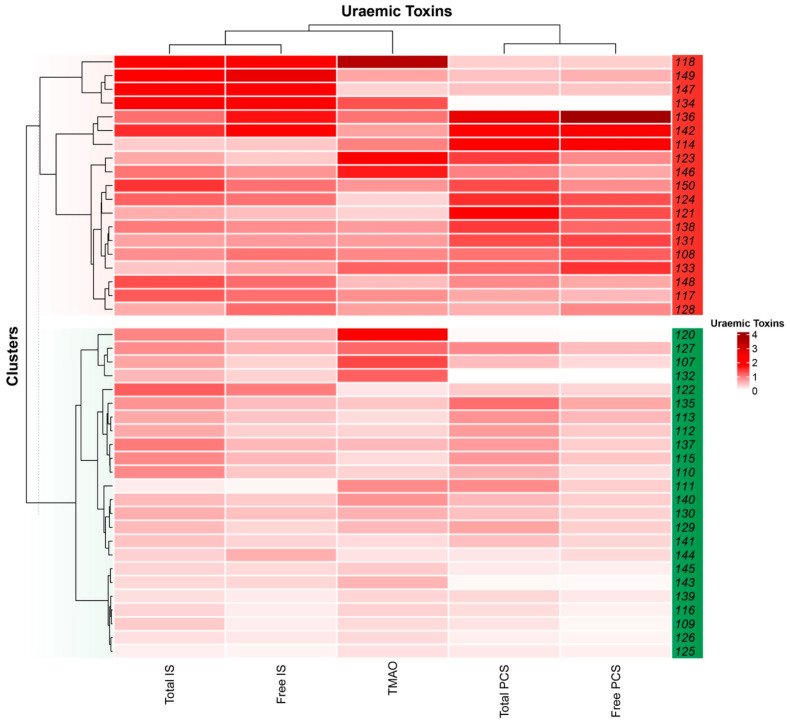
Uraemic toxin clusters. Clustering participants (*n* = 43) based on their baseline plasma uraemic toxin levels revealed two distinct groups. One cluster, termed the “high uraemic toxin group” (red; *n* = 18), exhibited elevated levels of uraemic toxins across all measured serum/plasma concentrations. In contrast, the other cluster, classified as the “low uraemic toxin group” (green, *n* = 26), displayed either negligible or lower concentrations of one or more uraemic toxins. In the low uraemic toxin group, mean serum free PCS = 2.1 (SD 1.3), total PCS = 69.3 (SD 46.1), free IS = 2.8 (SD 1.5), total IS = 40.6 (SD 20.6) and TMAO = 47.7 (SD 40.9). The high uraemic toxin group had mean serum free PCS = 9.0 (SD 6.8), total PCS = 176.3 (SD 104.7), free IS = 9.2 (SD 5.1), total IS = 84.5 (SD 42.4) and TMAO = 81.4 (SD 59.3).

**Figure 3 toxins-17-00334-f003:**
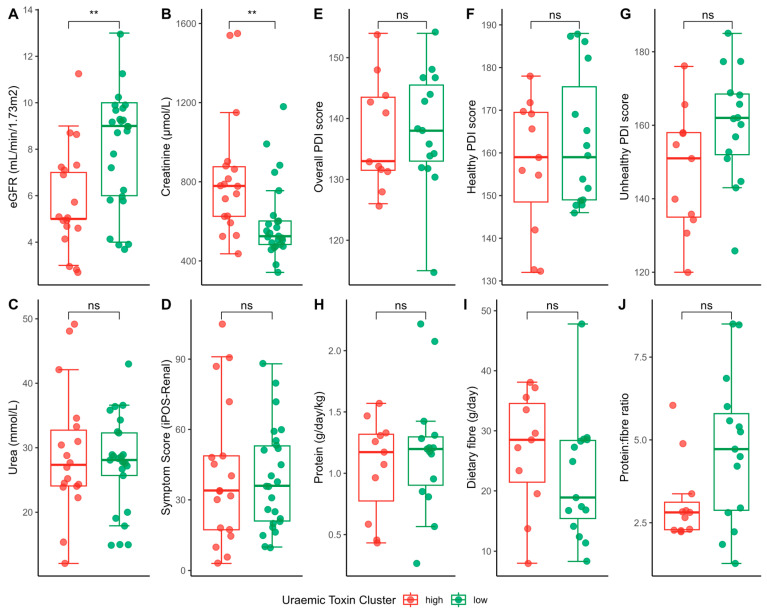
Differences in selected outcomes between uraemic toxin cluster groups. Clinical outcomes, including biochemical markers such as (**A**) eGFR, (**B**) creatinine, (**C**) urea, and (**D**) total participant-reported symptom scores. Dietary outcomes, including the (**E**) overall plant-based diet quality index [PDI], (**F**) healthy PDI, (**G**) unhealthy PDI, (**H**) protein and (**I**) fibre intake, and their (**J**) ratio. Baseline comparisons between these groups revealed participants in the high group had significantly worse kidney function compared to those in the low group (mean eGFR = 5.89 (SD 2.22) vs. 8.04 (SD 2.44) mL/min/1.73^2^, *p* < 0.01. Statistical differences were determined using either the Wilcoxon test or *t*-test, depending on normality and assumptions. ns = not statistically significant; ** *p*-value < 0.01.

**Figure 4 toxins-17-00334-f004:**
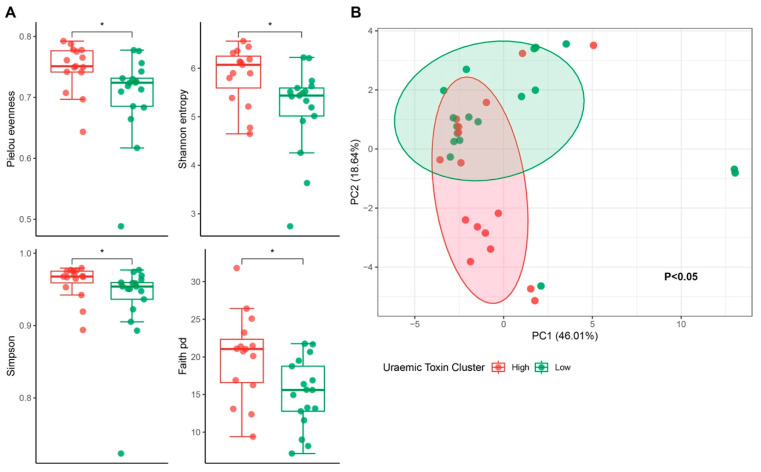
Differences in microbial diversity outcomes between uraemic toxin cluster groups. (**A**) Differences in alpha-diversity metrics, assessed using the Wilcoxon test (Mann–Whitney U test) (* *p* < 0.05). (**B**) PCA plot illustrating differences in microbiota profiles between cluster groups. Analysis is based on relative abundance data at the genus level that was arcsine square-root-transformed. Dissimilarities between samples using the Bray–Curtis dissimilarity index. Relative abundance filtering required taxa to be present in at least 20% of the total samples. *p*-value was derived from a Permutational Multivariate Analysis of Variance (PERMANOVA), adjusting for age, gender, and diabetes, using 999 permutations. Analysis was completed on available data from *N* = 34 participants.

**Figure 5 toxins-17-00334-f005:**
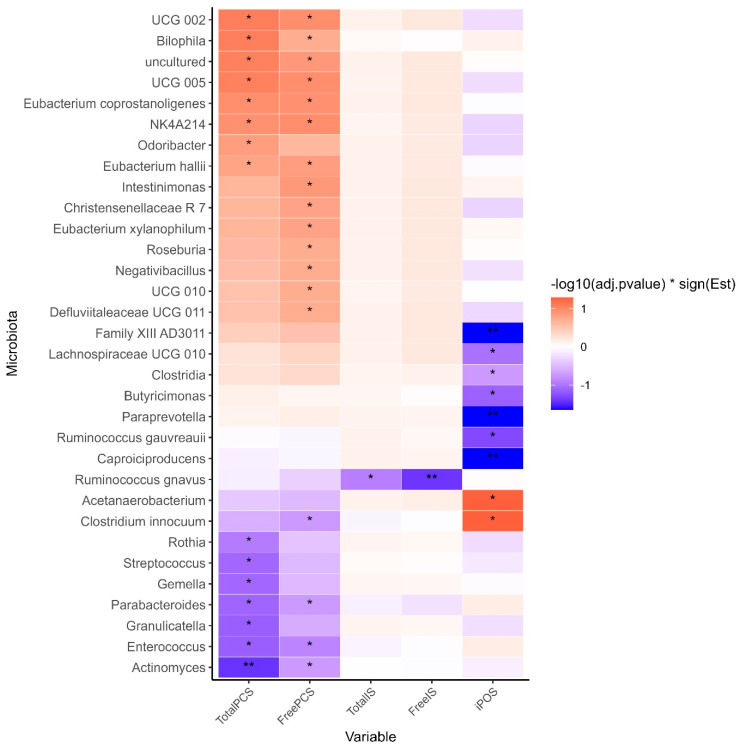
Associations between microbial taxa (genus level) and plasma uraemic toxin concentrations and total iPOS-renal symptom score. * Adjusted *p*-value < 0.20, ** adjusted *p*-value < 0.05. *p*-values were derived from linear regression models, where uraemic toxin concentrations and iPOS-renal symptom scores were log-transformed prior to analysis. Models were adjusted for age, gender, diabetes status, and blood albumin levels. Taxa relative abundances were arcsine square-root transformed and filtered to include only taxa present in at least 20% of samples prior to analysis. Analysis was conducted based on completed data from *N* = 34 participants data. PCS, p-Cresyl sulfate; IS, Indoxyl sulfate; iPOS, Integrated Palliative Care Outcomes Scale.

**Table 1 toxins-17-00334-t001:** Patient Characteristics.

Baseline Demographic Data		
	N	%
Sex		
Male	30	70
Female	13	30
Age (yrs)		
21–40	5	11.6
41–60	12	27.9
61–80	26	60.4
Primary Renal Disease		
Diabetes mellitus	22	51.2
Hypertension	12	27.9
ADPCKD	3	7.0
Glomerulonephritis	10	23.3
Other	8	18.6
Multiple	10	23.3
Comorbidities		
Hypertension	35	81.4
Ischemic Heart Disease	14	32.6
Diabetes mellitus	25	58.1
Hypercholesterolaemia	18	41.9
Gout	13	30.2
Antibiotics in the Previous 12 Months		
Yes	38	88.4
No	5	11.6
Cephalosporin	37	86.0
Penicillin	14	32.6
Sulfamethoxazole-Trimethoprim	4	9.3
Ciprofloxacin	3	7.0
Doxycycline	3	7.0
Metronidazole	3	7.0
Vancomycin	2	4.7
Regular Medications		
Total (median, IQR)	11	8–15
Proton pump inhibitor	17	38.6
Calcium-based phosphate binder	27	62.8
Non-calcium phosphate binder	11	25
Erythropoietin	25	58.1
Serum Biochemistry at Commencement of PD		
Creatinine (Median, IQR)	599	512–802
Urea (Median, IQR)	28	24–32
Serum Uraemic Toxin Concentration (Median, IQR)		
TMAO (RR: 1.28–19.67 µmol/L) [23]	48.0	22.8–73.3
Total IS (RR: 0.70–6.30 µmol/L) [21]	49.0	26.5–71.5
Free IS (RR: 0.0–0.19 µmol/L) [21]	3.9	1.3–6.6
Total PCS (RR: 0.0–38.4 µmol/L) [21]	97.0	39.5–154.5
Free PCS (RR: 0.14–2.44 µmol/L) [21]	3.1	0.8–5.5
Weight (kg)	78	67–91
BMI (kg/m^2^)	27	23–31

ADPCKD: autosomal dominant polycystic kidney disease; IQR: interquartile range; N: number of participants; PD: peritoneal dialysis; PCS: p-Cresyl sulfate; IS: Indoxyl sulfate; TMAO: Trimethylamine oxide; RR: reference range.

**Table 2 toxins-17-00334-t002:** Baseline dietary data.

	Median (IQR)	Reference Range [22,24]
Macronutrients		
Energy (kJ/day)	8017 (6404–9421)	
Energy (kJ/kg/day)	112 (80–127)	9500–12,100
Protein (g/day)	83 (64–103)	125–159
Protein (g/kg/day)	1.2 (0.9–1.3)	52–64
Fat (g/day)	79 (50–92)	0.7–0.8
Saturated fat (g/day)	22 (17–38)	
Carbohydrate (g/day)	180 (148–260)	
Micronutrients		
Phosphorus (mg)	1237 (1075–1585)	<1000
Potassium (mg)	2703 (2388–3469)	2800–3800
Sodium (mg)	1724 (1120–2440)	<2300
Calcium (mg)	638 (408–925)	1000–1300
Zinc (mg)	9 (6–12)	8–14
Vitamin B6 (mg)	1.4 (1.0–2)	1.5–1.7
Vitamin C (mg)	88 (44–136)	45
Fibre (g)	25 (17–29)	25–35
% kJ from fibre	2.2 (1.7–2.8)	
% kJ from carbohydrate	43 (39–49)	45–65
Core Food Groups		
Grain serves	6 (4–8)	4–6
Fruit serves	0.9 (0.5–1.8)	2
Vegetables serves	4 (2–5)	5.0–5.5
Dairy serves	1.1 (0.6–1.5)	≤1
Meat/alternative serves	2.5 (1.6–2.9)	2.0–2.5
Alcohol standard serves	0 (0–0)	<2
Added sugar (g)	15 (4–34)	<36
Diet Quality Indices (Range of possible scores: PDI: 46–230; Healthy PDI: 53–265; Unhealthy PDI: 51–225)		
Plant-Based Diet Index	138 (10)	
Healthy PDI	160 (14)	
Unhealthy PDI	155 (16)	

PDI: plant-based diet index.

**Table 3 toxins-17-00334-t003:** Differences in taxa abundance between high and low uraemic toxin cluster groups at baseline (genus level).

Genera	Log_2_ Fold Change	SE	*p*-Value	Adjusted *p*-Value
*Catenibacterium*	9.43	2.08	<0.01	<0.01
*Prevotella*	6.20	1.46	<0.01	<0.01
*Clostridia*	7.81	2.25	<0.01	0.02
*Lachnospiraceae UCG-004*	4.05	1.22	<0.01	0.03
*Christensenellaceae R-7*	3.43	1.14	<0.01	0.07
*Ruminococcus gnavus*	−3.83	1.35	<0.01	0.09
*Eubacterium coprostanoligenes*	3.32	1.22	0.01	0.11

## Data Availability

The original contributions presented in this study are included in this article and Appendix A. Further inquiries can be directed to the corresponding author.

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
