# Peer review of "Associations Between Uraemic Toxins and Gut Microbiota in Adults Initiating Peritoneal Dialysis"

_toxins, 2025, doi:10.3390/toxins17070334_

Round 1

Reviewer 1 Report

Comments and Suggestions for Authors

To the authors,

Thank you for the opportunity to review the manuscript. This study investigates the associations between gut microbiota, uremia, and uremic toxins in patients initiating peritoneal dialysis (PD). The impact of uremic toxins in PD patients is a topic of considerable interest to many clinicians, and the study is of clear significance. However, I have several concerns regarding the current manuscript, as outlined below.

Although the study focuses on PD patients, it is a cross-sectional analysis conducted within three days of PD initiation. It remains unclear whether the observed findings reflect the influence of PD itself. Additionally, the specific PD regimen used in these patients is not described. Under such early conditions, the findings may not differ substantially from those observed in non-dialysis patients. While the title emphasizes PD, it is uncertain whether the results can be generalized to the broader PD population. Without data from before PD initiation or from a later time point after PD initiation, it is difficult to determine at which stage of kidney failure these findings are applicable.

Furthermore, as this is a cross-sectional study, it is not possible to determine whether the observed alterations in gut microbiota are a cause or a consequence of kidney failure. As mentioned above, without data from both before and after the initiation of PD, the significance of the current findings remains limited.

Author Response

Thank you for the opportunity to review the manuscript. This study investigates the associations between gut microbiota, uraemia, and uremic toxins in patients initiating peritoneal dialysis (PD). The impact of uremic toxins in PD patients is a topic of considerable interest to many clinicians, and the study is of clear significance. However, I have several concerns regarding the current manuscript, as outlined below.

Although the study focuses on PD patients, it is a cross-sectional analysis conducted within three days of PD initiation. It remains unclear whether the observed findings reflect the influence of PD itself. Additionally, the specific PD regimen used in these patients is not described. Under such early conditions, the findings may not differ substantially from those observed in non-dialysis patients. While the title emphasizes PD, it is uncertain whether the results can be generalized to the broader PD population. Without data from before PD initiation or from a later time point after PD initiation, it is difficult to determine at which stage of kidney failure these findings are applicable.

We thank the reviewer for their comments. We agree that this study does not address the influence of peritoneal dialysis – it is a baseline analysis of patients as part of a longitudinal study where patient’s microbiome was analysed after 1 year of being on peritoneal dialysis. This is noted in the abstract and in the title which notes this patient cohort is at the initiation of peritoneal dialysis. 

Furthermore, as this is a cross-sectional study, it is not possible to determine whether the observed alterations in gut microbiota are a cause or a consequence of kidney failure. As mentioned above, without data from both before and after the initiation of PD, the significance of the current findings remains limited.

Thank you for this suggestion. We agree with the reviewer that this is an observational study which can only identify association and not establish causation. We have further emphasised this point in the limitations to make this clearer for readers.

Reviewer 2 Report

Comments and Suggestions for Authors

In this work, the authors reveal associations between uraemic toxins and microbiome when commencing peritoneal dialysis. Several suggestions are made as follows to improve the quality of the manuscript.

  1. The title makes no sense. Please change the title.
  2. All abbreviations should be substantiated for the first time.
  3. The abstract should be improved. Please more result in details in abstract.
  4. Increasing studies have demonstrated that the altered composition and function of gut microbiota were involved in CKD patients reported by the latest publications such as altered intestinal microbial flora and metabolism in patients with idiopathic membranous nephropathy, targeting Lactobacillus johnsonii to reverse chronic kidney disease, Lactobacillus species ameliorate MN through inhibiting the aryl hydrocarbon receptor pathway via tryptophan-produced indole metabolites, the role of the intestinal microbiome in cognitive decline in patients with kidney disease. The authors described that individuals with CKD have been found to have distinct gut microbiota composition and function. The reviewer suggests summarizing these latest studies form CKD patients.
  5. Please provide both manufacturer’s name and location (city, state, and country) for important equipment and reagents in the manuscript.
  6. More multivariate statistical analysis should be performed.
  7. Publishing clinical studies should be further discussed.
  8. The authors reveal associations between uraemic toxins and microbiome when commencing peritoneal dialysis. Increasing publications have shown that a variety of uremic toxins were identified in CKD. recently, a number of metabolites including uremic toxins identified from CKD patients were demonstrated to mediate renal fibrosis reported by recent and latest publications such as Cell Mol Life Sci 2024,81(1):480; Integrative Medicine in Nephrology and Andrology 11(3):e24-00002, September 2024; new insights into a novel metabolic biomarker and therapeutic target for chronic kidney disease; Clin Nutr ESPEN 2025 Feb:65:105-114. These studies uncovered that the associations between renal function and circulating metabolites including uremic toxins in CKD patients. Please discuss these studies to improve manuscript.
  9. Limitations should be described.
  10. The most of references are old. Please cite and update the latest publications.
  11. Please change the references based on the guide for authors.
  12. The language editing should be improved by a native speaker.

Author Response

Reviewer 2

  1. The title makes no sense. Please change the title.

We thank the reviewer for their comment. The previous title was:

Associations between uraemic toxins and microbiome when commencing peritoneal dialysis

Current title is: Associations between uraemic toxins and gut microbiota in adults initiating peritoneal .

We recognise that this is over the word limit for titles but given the reviewer’s comment on the lack of clarity of the previous title we request your consideration of this updated title.

  1. All abbreviations should be substantiated for the first time.

We thank the reviewer for drawing our attention to this oversight, this has since been corrected.

  1. The abstract should be improved. Please more result in details in abstract.

We thank the reviewer for their comment. The previous abstract was:

Declining kidney function contributes to accumulation of uraemic toxins produced by gut microbiota, leading to the uraemic syndrome. This study aimed to identify associations between uraemic toxins, symptoms and the gut microbiota in individuals commencing peritoneal dialysis. Baseline data of forty-three adults participating in a longitudinal observational study of the microbiome was analysed. Participants had a mean age of 61 years (SD = 13), median eGFR 7 ml/min/1.73 m2 (range 3–13), 70% were male and the primary cause of chronic kidney disease was diabetes mellitus in 51.2%. Symptom scores using the Integrated Palliative care Outcomes Scale-Renal were recorded. Plasma p-Cresyl sulfate, indoxyl sulfate and trimethylamine N-oxide were measured using liquid chromatography-mass spectrometry. Gut microbiota was determined using 16S rRNA sequencing. Associations between variables and relative abundances at the genus level were tested using multivariate associations with linear models. Patients were classified into ‘high’ and ‘low’ uraemic toxin groups with the high group having a lower eGFR (p<0.05). There was significant difference in alpha diversity (pFaith’s=0.01; pShannon=0.01) between these two groups as well as beta diversity (pPERMANOVA=0.01). Individual uraemic toxins particularly p-Cresyl sulfate were associated with increased abundance of taxa including UCG002 and reduction of Actinomyces. Patients with kidney failure commencing peritoneal dialysis have distinct uraemic toxin profiles, associated with differences in microbial diversity. This phenotype was also associated with differences in residual kidney function but not associated with diet or symptom severity. Future studies are required to validate these findings to inform future therapeutic strategies.

The current abstracts includes more results as requested:

Declining kidney function contributes to accumulation of uraemic toxins produced by gut microbiota, leading to the uraemic syndrome. This study aimed to identify associations between uraemic toxins, diet quality, symptoms and the gut microbiota in individuals initiating peritoneal dialysis. Cross sectional analysis of baseline data of participants in a longitudinal study was conducted. Symptom scores using the Integrated Palliative care Outcomes Scale-Renal were recorded. Plasma p-Cresyl sulfate, indoxyl sulfate and trimethylamine N-oxide were measured using liquid chromatography-mass spectrometry. Gut microbiota was determined using 16S rRNA sequencing. Multivariate linear models examined associations across the cohort.

Data from 43 participants (mean age 61 ± 13 years; 70% male; median eGFR 7 ml/min/1.73 m²) were analysed. Diabetes was the primary cause of kidney disease (51.2%). Patients were classified into ‘high’ (n=18) and ‘low’ (n=26) uraemic toxin groups using K-means clustering. The ‘high’ group had a lower eGFR (p<0.05) but no differences in diet quality or symptom scores. Significant differences in alpha and beta diversity were observed between both groups (p=0.01). The ‘high’ group had increased Catenibacterium, Prevotella, Clostridia, and decreased Ruminococcus gnavus abundances. Multivariate models identified 32 genera associated with uraemic toxins, including positive associations of Oscillospiraceae UCG-002 and UCG-005 with p-cresyl sulfate, and negative associations with Actinomyces and Enterococcus.

Patients with kidney failure initiating peritoneal dialysis have distinct uraemic toxin profiles, associated with differences in microbial diversity. This phenotype was also associated with differences in residual kidney function but not associated with diet or symptom severity. Longitudinal studies are required to determine causality and guide therapeutic interventions.

  1. Increasing studies have demonstrated that the altered composition and function of gut microbiota were involved in CKD patients reported by the latest publications such as altered intestinal microbial flora and metabolism in patients with idiopathic membranous nephropathy, targeting Lactobacillus johnsonii to reverse chronic kidney disease, Lactobacillus species ameliorate MN through inhibiting the aryl hydrocarbon receptor pathway via tryptophan-produced indole metabolites, the role of the intestinal microbiome in cognitive decline in patients with kidney disease. The authors described that individuals with CKD have been found to have distinct gut microbiota composition and function. The reviewer suggests summarizing these latest studies form CKD patients.

We have included a paragraph on these works as suggested by the reviewer at line 240-247.

With regard to specific taxa, previous studies have shown reduced abundance of genus Prevotella in CKD [33], but our data shows a significant difference in this genus between the high and low clusters. This builds on previous data showing correlations between specific bacteria, uraemic toxins and CKD progression. Faceal samples of CKD patients demonstrate lower relative abundance of short-chain fatty acid generating bacteria, such as Bifidobacterium spp., and higher relative abundance of Enterobacteriaceae in CKD [34]. In rat models, Fusobacterium nucleatum and Eggerthella lenta increase uraemic toxin production [34].  

  1. Please provide both manufacturer’s name and location (city, state, and country) for important equipment and reagents in the manuscript.

This has been included – thank you for noting this , for example at line 394.

  1. More multivariate statistical analysis should be performed.

We are unclear what additional analyses the reviewer would like to see beyond those that relate to the aims and objectives of the study. As a result, no changes have been made.

  1. Publishing clinical studies should be further discussed.

We assume that the reviewed suggests that published clinical studies on pre- pro and synbiotics should be further discussed in the discussion and we have included a further sentence including an updated meta-analysis at line 298

The authors reveal associations between uraemic toxins and microbiome when commencing peritoneal dialysis. Increasing publications have shown that a variety of uremic toxins were identified in CKD. recently, a number of metabolites including uremic toxins identified from CKD patients were demonstrated to mediate renal fibrosis reported by recent and latest publications such as Cell Mol Life Sci 2024,81(1):480; Integrative Medicine in Nephrology and Andrology 11(3):e24-00002, September 2024; new insights into a novel metabolic biomarker and therapeutic target for chronic kidney disease; Clin Nutr ESPEN 2025 Feb:65:105-114. These studies uncovered that the associations between renal function and circulating metabolites including uremic toxins in CKD patients. Please discuss these studies to improve manuscript.

We have included these studies as suggested at line 246.

  1. Limitations should be described.

Limitations are described in the paragraph starting at line 350. “ A limitation of this study is its cross-sectional design, which restricts the ability to infer causal relationships and limits the scope to associations only…”

  1. The most of references are old. Please cite and update the latest .

We have updated the references, for example at line 607.

  1. Please change the references based on the guide for authors.

We have updated the referencing format.

  1. The language editing should be improved by a native speaker.

We are native English speakers but have improved the clarity of the manuscript with the assistance of the reviewers’ comments.

Reviewer 3 Report

Comments and Suggestions for Authors

Line 17, the number of patients in each group (high and low uraemic toxin) has to be identified.

Please insert the full names of all abbreviations at their first mentions through the manuscript (e.g., SD in line 17,…..).

Line 19, for the results part in the abstract, please insert only the P significance values (P<0.05), you don’t need to insert the analysis (e.g., Faith’s=0.01; pShannon=0.01,….).

Line 43, please insert specific examples of the dysbiosis which related to the CKD, for example, insert names of the most common bacterial strains related to this disorder.

Line 355, how many patients (in total) were you used here, please provide more information about these patients, e.,g their age, gender,…

Line 363, when (Fasting or what?) and how did you collect the blood samples? you collected plasma so please mention the anticoagulant that you used here.   Also please insert the model, manufacturer, city, and country of the centrifuge that you used and any instruments used in this study in the manuscript.

Line 364, what do you mean by “Blood specimens were processed within four hours”, What this time is needed for?

I wonder these 24 h if the samples were collected from home, these 24h may affect the microbial composition of the collected stool. Please clarify this point.

Line 395, please add the model, city and country of the spectrophotometer that you used, please do the same for any instruments that you used also for the kits, LCMS,…

Line 404, needs English editing.

Line 404, Briefly please insert the separation conditions in the LCMS.

The division of groups are not clear in the methods part!

The Uraemic Toxin Analysis and Statistical Analysis are the basic of this work and ca not be considered as Supplementary methods!

The first part of the results (Participant Characteristics and  Uraemic Symptoms 99) as well as tables number 1 and 2, and Fig 1 actually related to the methods part nor the results, your results have to be started when you divided the patients into subgroups.

In the results part you don’t need to insert the values again that mentioned in the tables or figures, just mention if there s increases or decreases in the experimental parameters among the subgroups.

Line 281, how much the concentration of these dietary fiber intakes of these studies?

Author Response

Reviewer 3

Line 17, the number of patients in each group (high and low uraemic toxin) has to be identified.

We thank you for identifying this oversight, we have corrected this now in the updated abstract. See lines 15-16: Participants were grouped into ‘high’ (n=18) and ‘low’ (n=26) uraemic toxin profiles using K-means clustering.

Please insert the full names of all abbreviations at their first mentions through the manuscript (e.g., SD in line 17,…..).

We thank you for identifying this oversight, all abbreviations have been defined and updated throughout the text.

Line 19, for the results part in the abstract, please insert only the P significance values (P<0.05), you don’t need to insert the analysis (e.g., Faith’s=0.01; pShannon=0.01,….).

We have now corrected this in the updated abstract at line 17.

Line 43, please insert specific examples of the dysbiosis which related to the CKD, for example, insert names of the most common bacterial strains related to this .

We have included this as suggested.

Line 355, how many patients (in total) were you used here, please provide more information about these patients, e.,g their age, gender,…

Thank you for this comment. To avoid redundancy within the word limit, we have not repeated this information in the Methods section, as it is already presented in the Results (lines 85-88).

Line 363, when (Fasting or what?) and how did you collect the blood samples? you collected plasma so please mention the anticoagulant that you used here.   Also please insert the model, manufacturer, city, and country of the centrifuge that you used and any instruments used in this study in the manuscript.

We have included this detail as requested.

Line 364, what do you mean by “Blood specimens were processed within four hours”, What this time is needed for?

We also collected peripheral blood mononuclear cells for future analysis but recognise that this detail may be confusing for this particular paper as these data are not included and thus have deleted it.

I wonder these 24 h if the samples were collected from home, these 24h may affect the microbial composition of the collected stool. Please clarify this point.

Thank you for this question. Preservation of DNA at the time of collection using Zymo DNA/RNA collection tubes is up to 2 years at room temperature and RNA is presented for 1 month at room temperature. This detail has now been included in the methods.

Line 395, please add the model, city and country of the spectrophotometer that you used, please do the same for any instruments that you used also for the kits, ,…

We have included this detail as requested.

Line 404, needs English editing.

We have addressed this typographical error

Plasma PCS and IS were measured using liquid chromatography-mass spectrometry (LCMS) as previously described [42].

Line 404, Briefly please insert the separation conditions in the LCMS.

This has now been included:

The division of groups are not clear in the methods part!

Thank you for this comment. We have included comprehensive details in the statistical analysis section of the methods. Please see lines 483 to 489:

“To identify subgroups with distinct serum uremic toxin concentrations, k-means clustering was applied to categorise participants based on their uremic toxin profiles. One participant was excluded due to lost plasma samples, resulting in a final analytical sample of 43 participants. Clustering was performed using the ComplexHeatmap package in R [50], where data were scaled but not centred (scale, center = FALSE). The clustering analysis used Euclidean distance for row-wise clustering and Ward’s method (ward.D) to define cluster membership”

No change was made.

The Uraemic Toxin Analysis and Statistical Analysis are the basic of this work and canot be considered as Supplementary methods!

We have included LCMS methodology in the methods.

We agree that the statistical analysis is key to this work and have provided a detailed summary in the methods section of the main manuscript. Please see lines 481 to 523.

The first part of the results (Participant Characteristics and Uraemic Symptoms 99) as well as tables number 1 and 2, and Fig 1 actually related to the methods part nor the results, your results have to be started when you divided the patients into .

We respectfully disagree and feel this is valid reporting of results. However we see the reviewer’s point and have included additional wording on subgroup analysis in the abstract at line 15. In this study we analysed associations as a whole cohort but we also divided the patients into subgroups using cluster analysis after the patients uraemic toxin levels were measured.

In the results part you don’t need to insert the values again that mentioned in the tables or figures, just mention if there s increases or decreases in the experimental parameters among the subgroups.

This was previously included for ease of reading but has been deleted as requested.

Line 281, how much the concentration of these dietary fiber intakes of these studies?

We have now included this detail at line 304

Reviewer 4 Report

Comments and Suggestions for Authors

The study titled "Associations between uraemic toxins and microbiome when commencing peritoneal dialysis" explored the associations between uraemic toxins, symptoms and the gut microbiota in individuals commencing peritoneal dialysis. The study is informative and the findings are interesting. I have the following questions and comments. 

1, the inclusion and exclusion criteria have not been specified in the manuscript. Please revise. 

2, in table 1, I noticed that 88.4% of the patient has received antibiotics in the past year. Could this affect the conclusion of the study? What kind of antibiotics were used? This should be specified and discussed. 

3, for all the tables, I suggest the authors to change them into three-line format. 

4, in figure 1, the title of the y axis should be "PATIENTS REPORTING SYMPTOM (%)".

5, in figure Figure 3, each panel should be labeled from A to Z. 

6, the limitations of the study must be clearly discussed. Please revise. 

Author Response

Reviewer 4

The study titled "Associations between uraemic toxins and microbiome when commencing peritoneal dialysis" explored the associations between uraemic toxins, symptoms and the gut microbiota in individuals commencing peritoneal dialysis. The study is informative and the findings are interesting. I have the following questions and comments. 

1, the inclusion and exclusion criteria have not been specified in the manuscript. Please revise. 

We have included this in the methods section 5.1.

2, in table 1, I noticed that 88.4% of the patient has received antibiotics in the past year. Could this affect the conclusion of the study? What kind of antibiotics were used? This should be specified and discussed. 

Thank you for this comment. We note that this is a limitation in the discussion. Antibiotic use in dialysis patients is pervasive and if this is an exclusion criterion only a very select group of patients is likely to be included. As other studies have not documented the presence/absence of antibiotic use in patients with chronic kidney disease it is difficult to compare with other works. We have included the type of antibiotics in Table 1. Additionally, we have added the statement (line 360), “While this poses challenges for microbiome-related investigations, it nonetheless reflects real-world clinical practice,” to the limitations section, as this consideration is relevant across many clinical contexts.

3, for all the tables, I suggest the authors to change them into three-line . 

Thank you for this comment. Tables have been changed to a consistent format throughout.

4, in figure 1, the title of the y axis should be "PATIENTS REPORTING SYMPTOM (%)".

We have corrected this oversight, thank you

5, in figure Figure 3, each panel should be labeled from A to Z. 

We have corrected this oversight thank you

6, the limitations of the study must be clearly discussed. Please revise. 

This section has been more clearly indicated in the manuscript at lines 352 – 365

Round 2

Reviewer 1 Report

Comments and Suggestions for Authors

To the Authors:

Thank you for the opportunity to review your revised manuscript. I appreciate your sincere effort in responding to the reviewers’ comments. However, the responses provided remain rather superficial and do not address the fundamental issues raised.

In particular, including "peritoneal dialysis" in the title is highly misleading to readers, given that the condition described is not influenced by peritoneal dialysis. Moreover, there remains a clear discrepancy between the stated aim of the study and the actual results. As in my previous review, I must reiterate that the findings of this study appear to be of limited significance.

Reviewer 2 Report

Comments and Suggestions for Authors

The authors have improved the manuscript. Therefore, I suggest that the manuscript is published.

Reviewer 3 Report

Comments and Suggestions for Authors

The authors modified the manuscript, and it can be accepted for publication.

Reviewer 4 Report

Comments and Suggestions for Authors

The authors have revised the manuscript accordingly. It can be considered for publication.